# Impact of COVID-19 on the awareness and interest in infectious disease specialization among Japanese medical students

Naruto Kamada[1], Hideharu Hagiya[2]*, Satoshi Kutsuna[3,4]

1 Faculty of Medicine, Wakayama Medical University, Wakayama, Japan, 2 Department of Infectious Diseases, Okayama University Hospital, Okayama, Japan, 3 Department of Infection Control and Prevention, Graduate School of Medicine, Osaka University, Osaka, Japan, 4 Division of Fostering Required Medical Human Resources, Center for Infectious Disease Education and Research (CiDER), Osaka University, Osaka, Japan

* hagiya@okayama-u.ac.jp

## Abstract

### Background

The SARS-CoV-2 pandemic has highlighted the critical deficiency of infectious disease (ID) specialists, a subspecialty that remains underrepresented among Japanese medical students.

### Methods

This nationwide cross-sectional survey was administered between April and August 2024 via an online questionnaire distributed to medical students throughout Japan. The survey assessed awareness of and interest in ID specialization, categorizing students by academic year: lower (first- and second-year students), middle (third- and fourth-year students), and upper grades (fifth- and sixth-year students).

### Results

Of 502 respondents, data for 492 medical students were eligible, of whom 69.7% demonstrated awareness of ID specialists, with recognition rates increasing proportionally with academic progression. Regarding career aspirations, 9.8% of respondents expressed interest in pursuing ID specialization, with the highest proportion observed among upper-grade students (19.4%). Male students (14.8%) expressed greater interest in ID specialization than female students (5.2%). The pandemic positively influenced 5.5% of students to consider ID specialization as a future career, whereas only 0.6% reported a negative impact.

**Data availability statement:** Raw dataset is deposited at Dryad: https://datadryad.org/dataset/doi:10.5061/dryad.r7sqv9sqw.

**Funding:** This work was conducted as part of "The Nippon Foundation - Osaka University Project for Infectious Disease Prevention". The funders had no role in study design, data collection and analysis, decision to publish, or preparation of the manuscript.

**Competing interests:** The authors have declared that no competing interests exist.

**Abbreviations:** CI, Confidence interval; COVID-19, Coronavirus disease 2019; ID, Infectious disease; OR, Odds ratio; SARS-CoV-2, Severe acute respiratory syndrome coronavirus 2.

## Conclusions

These findings underscore the necessity of enhanced educational initiatives to promote ID specialization among medical students, addressing current shortages and future infectious disease preparedness.

---

## Introduction

Following its first report in Wuhan, China, severe acute respiratory syndrome coronavirus 2 (SARS-CoV-2), the causative pathogen of the novel coronavirus disease 2019 (COVID-19), rapidly disseminated globally, sparking an unprecedented pandemic [1,2]. Despite extensive global efforts to underscore the importance of infection control and prevention strategies, SARS-CoV-2 has repeatedly triggered widespread outbreaks and resurgences [3]. Healthcare systems worldwide had to adapt to diverse challenges and demands to secure public health and sustain essential societal functions. The situation was further intensified by the recurrent emergence of genetic variants that evade pre-existing immunity at both individual and population levels, necessitating swift and flexible approaches to mitigate this ongoing infectious crisis [4].

During the pandemic, infectious disease (ID) specialists exerted clinical and administrative leadership by leveraging their extensive knowledge and expertise in infection control, prevention, diagnosis, and treatment. Indeed, many ID specialists demonstrated exceptional capabilities and outperformed in both patient care in hospital settings as well as public management at regional or national levels. However, the number of ID specialists in Japan remains low, even in the aftermath of the pandemic; as of April 15, 2025, only 1,876 medical doctors were certified as ID specialists by the Japanese Society of Infectious Diseases [5]. This shortage is insufficient to meet the growing demand [6], highlighting the need for increased educational and training opportunities for medical students and early-career physicians. A similar shortage of ID specialists is evident in the United States, where 93 of 189 (49.2%) residency programs remained unfilled in 2024 [7]. As the global threat of emerging and re-emerging infectious diseases intensifies, the demand for ID specialists continues to increase. The risks associated with infectious diseases escalate with globalization, further amplifying the need for the expertise of ID specialists.

A significant contributing factor to the shortage of ID specialists could be attributed to a lack of awareness of this specialty among medical students. A 2021 study reported notably low interest in ID specialization among medical students [8], with some students being dissuaded from pursuing careers in this field due to their experiences during the pandemic. However, given that this study was limited to a single medical school, its findings may not accurately represent broader trends. Consequently, the present survey aims to evaluate Japanese medical students' current awareness of and interest in ID specialization.

## Participants and methods

### Study design

This nationwide cross-sectional descriptive study was conducted between April 12 and August 31, 2024. A web-based questionnaire created using Google Forms was distributed to medical students by the researchers through the aid of the Japan Association for Medical Student Societies, a voluntary membership organization mainly run by medical students themselves. Distribution methodologies were institution-specific, depending on the members of each institution. At the time of study implementation, Japan's medical education landscape comprised 82 medical schools (42 national, 8 public, 31 private, and 1 national defense), with a collective enrollment of approximately 56,000 medical students. Informed consent was obtained from each participant before the completion of the questionnaire. The study was approved by the review board of Okayama University Hospital (approval No. 2312−036). No specific exclusion criteria were established, and responses were collected anonymously.

### Definition of student groups

Considering the impact of the COVID-19 pandemic, which began in January 2020, we categorized students by academic year as follows: lower grades consisted of first- and second-year students, the majority of whom were considered to have experienced the pandemic during their third year of junior high school and first year of high school; middle grades included third- and fourth-year students who were mostly affected by the pandemic during their second and third years of high school; and upper grades comprised fifth- and sixth-year students, many of whom experienced the pandemic after entering university. Lower-grade students had not yet attended any clinical lectures as part of their medical school curriculum. Most middle-grade students had begun attending clinical lectures on infectious diseases but had not yet started their clinical clerkships. Upper-grade students had begun their clinical clerkships.

### Questionaries

The web-based survey included the following questions: "Are you aware of the existence of ID specialists?," "(If so) Were you aware of the existence of ID specialists before the COVID-19 pandemic?," "Have you ever been interested in pursuing a career as an ID specialist?," "Have your impressions of ID specialists changed after witnessing the COVID-19 pandemic?," "Were you previously, and are you currently, interested in becoming an ID specialist?," and "Did the COVID-19 pandemic motivate your decision to attend medical school?" The final question was directed only at low- and middle-grade students. The questionnaire items were methodically formulated and refined through discussions among the principal investigators of the study.

### Statistical analysis

Categorical variables are presented as numbers, percentages, and odds ratios (OR) with 95% confidence intervals (CIs), assessed using the chi-square test. The data were analyzed using EZR software, a graphic user interface for the R 4.3.1 software (The R Foundation for Statistical Computing, Vienna, Austria). All estimates were expressed as point estimates with 95% CI, and all reported p-values$<0.05$ were considered statistically significant.

## Results

A total of 502 respondents completed the survey. Among them, 492 (237 males, 250 females, and five who did not respond) from 29 medical universities consented to participate. The number and proportion of participants are shown in Table 1. The distribution of respondents' affiliated universities by academic year is summarized in S1 Table.

**Table 1. Numbers and proportions of the participants, by academic years and sex.**

| | Academic years | | | | | | Total |
| | 1st | 2nd | 3rd | 4th | 5th | 6th | |
|---|---|---|---|---|---|---|---|
| Male | 26 (30.6) | 23 (46.0) | 103 (54.5) | 26 (42.3) | 29 (55.8) | 30 (53.6) | 237 (48.2) |
| Female | 58 (68.2) | 24 (48.0) | 85 (45.0) | 34 (56.7) | 23 (44.2) | 26 (46.4) | 250 (50.8) |
| No response | 1 (1.2) | 3 (6.0) | 1 (0.5) | 0 | 0 | 0 | 5 (1.0) |
| Total | 85 | 50 | 189 | 60 | 52 | 56 | 492 |

Percentages are denoted in parentheses

## Awareness of ID specialty

Among the respondents, 343 (69.7%) reported having heard of ID specialists (**Fig 1**). The awareness of ID specialists tended to increase with academic year, from 56.3% in the lower grades to 87.0% in the upper grades, differing with statistical significance ($p = 0.014$ for lower vs. middle grades; $p < 0.001$ for middle vs. upper grades). Additionally, 103 medical students (20.9%) reported that they were aware of the existence of ID specialists before the COVID-19 pandemic. Prior awareness also significantly increased with academic year, with awareness rates of 12.6%, 19.7%, and 34.3% in the lower, middle, and upper grades, respectively.

This figure displays the proportions of respondents who answered "Yes" to the following questions: "Are you aware of the existence of ID specialists?" and "(if so) Were you aware of the existence of ID specialists before the COVID-19 pandemic?".

(A) The actual numbers of responses are as follows: 76/135 in Lower grades, 173/249 in Middle grades, 94/108 in Upper grades, and 343/492 in total.

(B) The actual numbers of responses are as follows: 17/135 in Lower grades, 49/249 in Middle grades, 37/108 in Upper grades, and 103/492 in total.

## Interest in becoming an ID specialist

Among the respondents, 48 (9.8%) expressed interest in pursuing a career as ID specialists (**Fig 2**). The proportion of interest in ID specialization was significantly higher among upper-grade students than among lower- or middle-grade students (19.4% vs. 6.7% and 7.2%, respectively). In terms of gender, the proportion of male students expressing interest was significantly higher than that of female students (14.8% vs. 5.2%; $p < 0.001$; OR: 3.2 [95% CI: 1.6–6.7]). Among those who indicated awareness of ID specialists (**Fig 1A**), 14.0% (48/343) reported interest in becoming ID specialists. This proportion was higher among upper grades than among lower and middle grades (22.3% vs. 11.8% and 10.4%, respectively).

This figure illustrates the proportion of respondents who answered "Yes" to the question: "Have you ever been interested in pursuing a career as an ID specialist?".

(A) Overall (N = 492). The actual numbers of responses are as follows: 9/135 in Lower grades, 18/249 in Middle grades, 21/108 in Upper grades, and 48/492 in total.

(B) By gender (N = 487, excluding 5 unknown). The actual numbers of responses are as follows: 13/250 in female and 35/237 in male, where the odds ratio for the difference was 3.2 (95% confidence interval: 1.6–6.7).

(C) Focusing on those who indicated awareness of ID specialists (N = 343). The actual numbers of responses are as follows: 9/76 in Lower grades, 18/173 in Middle grades, 21/94 in Upper grades, and 48/343 in total.

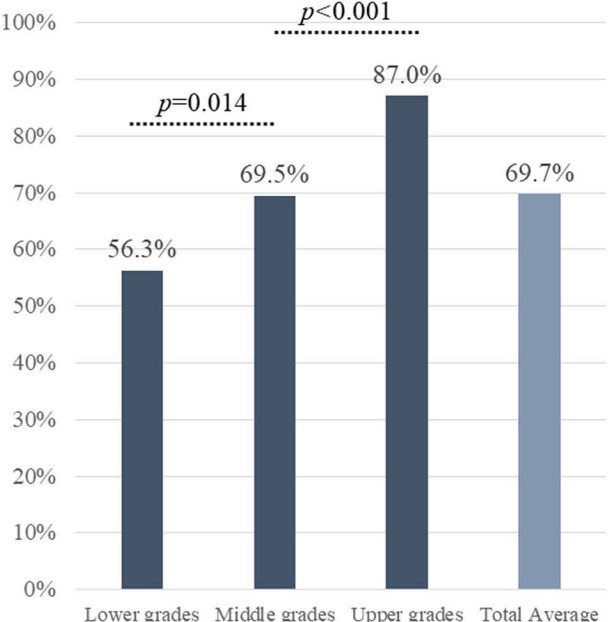
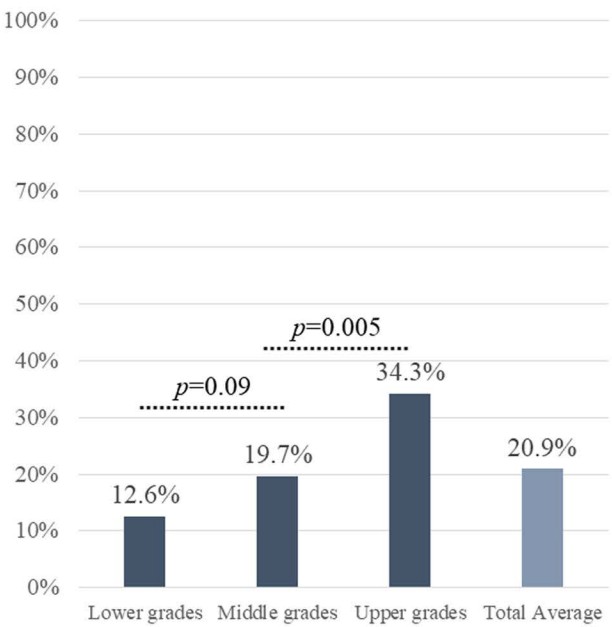

(A) Are you aware of the existence of ID specialists?

(B) (If so) Were you aware of the existence of ID specialists before the COVID-19 pandemic?

**Fig 1. Awareness of infectious disease (ID) specialists among 492 medical students across Japan.**

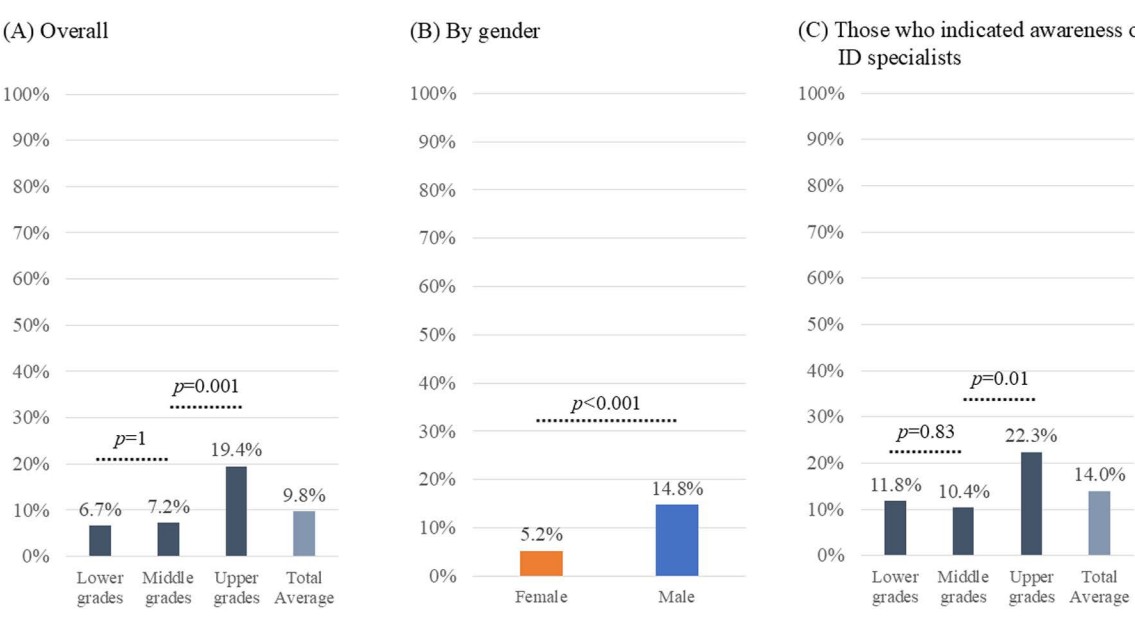

(A) Overall

(B) By gender

(C) Those who indicated awareness of ID specialists

**Fig 2. Interest in becoming an ID specialist.**

## Impact of COVID-19 on interest in becoming an ID specialist

Fig 3 summarizes responses regarding the impact of the COVID-19 pandemic on perceptions of ID specialization as a potential career path. Overall, 64.0% (315/492) respondents indicated that their pandemic experience did not influence their interest in pursuing a career as an ID specialist, and no significant variation was observed among academic grades. A total of 3.5% (17 students; two in the lower grades, seven in the middle grades, and eight in the upper grades) reported maintaining their interest in becoming ID specialists throughout the pandemic, while 5.5% (27 students; nine each in lower, middle, and upper grades) noted a shift in their interest toward ID specialization following the pandemic. Conversely, 0.6% (three students; one in the lower grades and two in the upper grades) indicated a decreased willingness to pursue a career as an ID specialist after the pandemic.

This figure summarizes the proportions of respondents who answered the questions (N = 492): "Have your impressions of ID specialists changed after witnessing the COVID-19 pandemic?" and "Were you previously, and are you currently, interested in becoming an ID specialist?". The actual numbers of responses in total are as follows: 315 in "No previously, and No currently", 17 in "Yes previously, and Yes currently", 27 in "No previously, but Yes currently", 3 in "Yes previously, but No currently", and 130 in "No idea".

## Impact of COVID-19 on interest in medical education

Finally, we inquired among lower- and middle-grade students about the influence of COVID-19 on their decision to pursue a medical education (Fig 4). Of the 382 medical students surveyed, 85.2% indicated that the pandemic did not affect their career choices. In contrast, 12.7% (49 students) reported that their experience with infectious hazards positively influenced their decision to pursue a medical career. Of these, 2.3% (nine students) stated that the COVID-19 pandemic

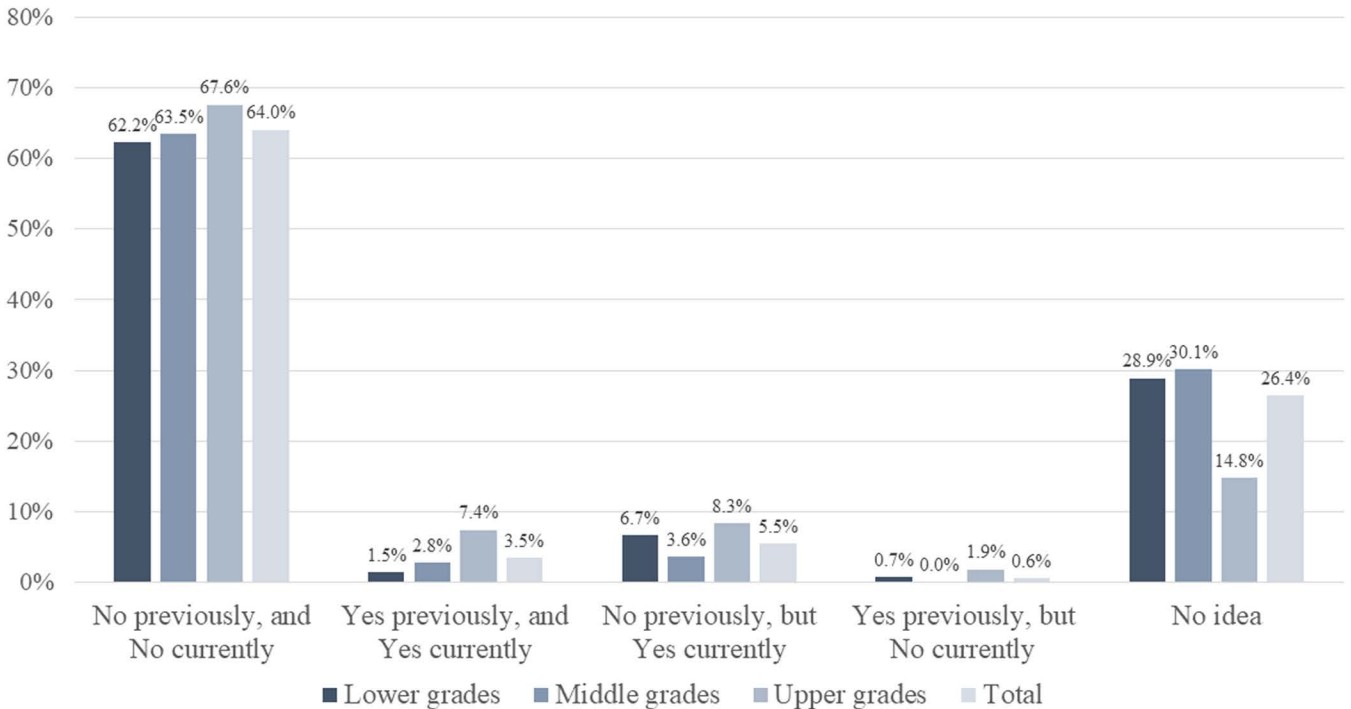

**Fig 3. Impact of the COVID-19 pandemic on the decision to pursue a career as an ID specialist.**

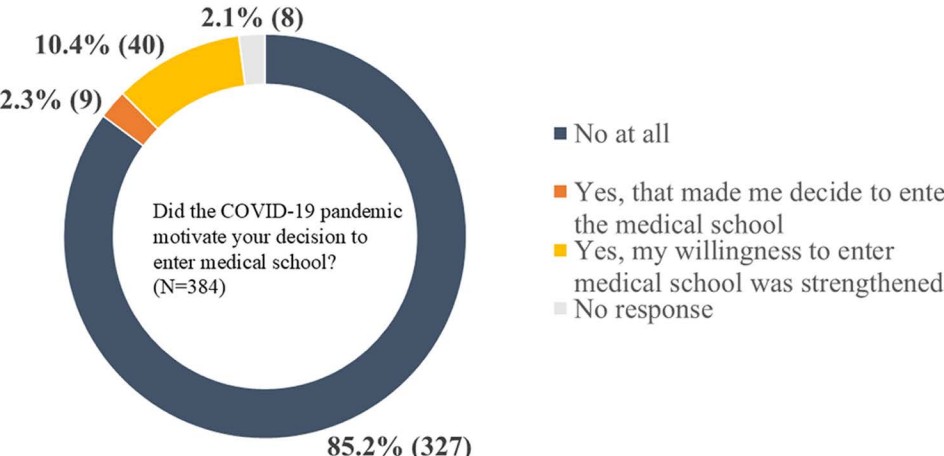

**Fig 4. Impact of COVID-19 pandemic on decisions to pursue admission to medical school.**

motivated them to enter medical school, while 10.4% (40 students) noted that their experiences during the pandemic strengthened their resolution to enroll in medical school.

This figure presents the proportion of respondents who answered the question, "Did the COVID-19 pandemic motivate your decision to attend medical school?". This question was directed solely at lower- and middle-grade students (N = 384). The actual numbers of responses are given in parentheses.

## Discussion

Through this questionnaire, we aimed to understand the awareness and interests of medical students in pursuing ID specialization as a potential career. Their overall awareness of ID specialists was 69.7%, which is consistent with data from a previous study (69.2%) [8]. The awareness level among upper grades was high (87.0%), aligning closely with earlier findings (84.6%–91.2%) [8]. However, only one-third of the respondents were aware of the existence of ID specialists before the COVID-19 pandemic, indicating that the global health crisis significantly heightened awareness of ID specialists among medical students.

The impact of the COVID-19 pandemic on students' decisions to pursue ID specialization as a career option is particularly noteworthy in this study. Data from a previous study indicated that 6.1% of students expressed an interest in ID specialization that had diminished post-pandemic, and 11.0% reported a reluctance to pursue this path as a consequence of their pandemic experiences [8]. Encouragingly, 5.5% of the medical students in this study reported that the pandemic newly motivated them to pursue careers as ID specialists, while 3.5% maintained their enthusiasm throughout this period. The pandemic negatively impacted the aspirations of only 0.6% of students. Additionally, 12.7% of students in the lower and middle grades indicated that the COVID-19 pandemic had positively influenced their decisions to attend medical school.

Historically, emerging diseases, including severe acute respiratory syndrome (SARS) [9,10], 2009 H1N1 influenza, and Middle East respiratory syndrome, have significantly influenced medical education. Similarly, COVID-19 has adversely affected both pre-clerkship and clerkship learning experiences [11–13]. Much like other social activities, medical education was compelled to transition from in-person instruction to online formats, and clinical practice was suspended at the majority of medical schools worldwide [14–16], raising concerns regarding the quality of medical education. Although remote learning has the potential to offer medical students increased effectiveness, accessibility, and flexibility in their studies [17], nearly half of students reported feeling unprepared for clinical practice because of the lack of hands-on training in essential clinical skills at medical schools [18].

This drastic shift in educational formats has likely posed significant challenges to students' career choices following graduation [19]. Results from an online questionnaire revealed that 66.6% (338/507) of Japanese medical students reported that the COVID-19 pandemic had adversely influenced their choice of training hospitals, with a noticeable trend toward avoiding internal medicine as a specialty [20]. A survey at Hokkaido University showed a decline in plans to pursue clinical and research training abroad among medical students following the pandemic [21]. In the United States, owing to the lack of opportunities to explore specialties of interest and secure recommendation letters for residency applications, approximately 20% of medical students reported negative impacts on their career choices [22]. In contrast, a study from China found that the willingness to pursue internal medicine as a future career remained relatively high (67%) even during the pandemic [23]. Due to the inherent heterogeneity in medical education system and sociocultural contexts across global regions, direct comparative analysis presents methodological challenges. However, considering that the overarching goal of medical education is to enhance the quality of healthcare provided to the public, this decline in educational quality could lead to long-term detrimental effects on public health [24].

Interest in ID specialization as a future career was unexpectedly high in this study, with 6.7% among the lower grades, 7.2% among the middle grades, and 19.4% among the upper grades. Overall, approximately one-tenth of medical students expressed an intention to pursue a career as an ID specialist in the future. While this may partly reflect a selection bias among respondents, it is an encouraging indicator of the potential increase in the number of ID specialists moving forward. Having accessible role models is crucial for undergraduate students in selecting their future specialties [25]. However, the reduction in opportunities for clinical practice during the pandemic has led to decreased exposure to role models, particularly in the field of infectious diseases [26]. Therefore, it is essential for ID specialists to actively present themselves as positive role models at every educational opportunity.

In general, ID specialists are expected to demonstrate expertise in the diagnosis and management of diverse infectious diseases. Additionally, they are better to possess comprehensive knowledge of infection prevention and control skills and experiences, demonstrate administrative leadership capabilities, and maintain proficiency in public health practices. However, current educational programs for ID specialists may not adequately address all these competencies. To cultivate well-rounded ID specialists who cover this comprehensive skill set, it is imperative to consider quality improvement measures within specialist education programs.

Our data also indicate that interest in ID specialization was notably higher among male students than among female students. Although the exact reason for this difference between gender is unclear, it may be attributed to the shortage of female ID specialists [6]. Unfortunately, female doctors generally tend to receive fewer clinical opportunities [27] and unfortunately occupy fewer leadership positions [28]. Nevertheless, they often outperform their male counterparts in various medical procedures, including central venous catheterization [29] and surgical operations [30]. To address the underappreciation of the ID specialty among female physicians, it is essential to highlight the importance and diversity of roles for ID specialists. Many ID specialists in hospitals primarily serve as consultants without the responsibilities typically associated with attending physicians, such as emergency calls or night shifts. This allows for a more manageable work-life balance, making it feasible for them to juggle personal responsibilities such as childcare alongside a full-time medical career. The issue of insufficient ID specialists can be partially alleviated by increasing female involvement in this field.

The etiological factors contributing to the limited professional appeal of ID specialization among medical students remain incompletely elucidated. While compensation disparities among medical subspecialties might influence career selection preferences, empirical facts from the Japanese healthcare system do not support this hypothesis as a contributory factor. Their quality of life would be comparatively favorable, considering the consultative nature of their practices that exempts after-hours clinical coverage responsibilities. Historically, opportunities and infrastructures for ID-specialized training were poor in Japan, thereby restricting access to the fellowship program for early-career physicians.[31] We postulate that this educational deficit has subsequently resulted in an absence of professional role models for medical students, perpetuating a negative feedback cycle that impedes recruitment into the ID specialty.

To foster the development of ID specialists, we advocate multifaceted approaches to enhance opportunities for ID education in medical schools. These could include designating ID specialists with recognized national credentials, establishing dedicated Departments of Infectious Diseases within medical and graduate schools, endowing educational courses for ID specialists funded by local governments, and recruiting young physicians engaged in healthcare services in remote areas to serve as ID specialists [31]. ID specialization is highly attractive and worth pursuing for students, given its nature of a multi-system approach, focus on the management of curable diseases, an integrative relationship with macro-epidemiology, and a global perspective [32].

The strength of this study lies in the collection of data from multiple facilities rather than from a single institution. However, the distribution methodologies varied according to individual institutions, thereby precluding accurate quantification of the total survey instruments distributed and the subsequent response rate. The total number of responses was limited to 492, representing only approximately 1% of medical students in Japan. Also, the disproportionate representation across academic years potentially introduced selection bias that may have influenced the results. Additionally, 106 responses (21.5%) were from a single medical school, which may have introduced bias into the results. A potential correlation might have existed between the presence of ID departments and student response patterns; however, adjustment for this variable was not feasible due to the absence of such demographic data in our data collection. Furthermore, only those with a particular interest in the subject participated in the survey, suggesting a potential selection bias. We speculate that non-responding students may have had little interest in the topic, indicating that negative perceptions of ID specialists may be significantly higher than the findings suggest. Finally, due to the inherent limitations of the cross-sectional study design, a causal relationship between the COVID-19 pandemic and the observed changes in students' career aspirations cannot be definitively established. A longitudinal cohort study design would have potentially yielded more robust conclusions regarding temporal associations and causal mechanisms. Nonetheless, we believe that this study provides valuable insights that can inform future educational and recruitment strategies for ID specialists, as similar data are not readily available.

In conclusion, our efforts revealed that the unprecedented experience of the pandemic did not necessarily adversely affect Japanese medical students' preferences for pursuing ID specialization as a future career. The interpretation of these results is constrained by selection bias and lack of representativeness, stemming from the heterogeneous distribution of respondent's universities and academic grades. However, awareness regarding the existence of ID specialists remains relatively low in this population. To effectively prepare for the next global pandemic, it is essential to educate and nurture those willing to become ID specialists. Future investigations should incorporate a multinational perspective, which would enhance our understanding of the relationship between pandemic experiences and medical students' career development, thereby contributing to the international academic community.

## Supporting information

**S1 Table. Breakdown list of the number of respondents from medical schools by student's grades.**
(DOCX)

## Author contributions

**Conceptualization:** Hideharu Hagiya, Satoshi Kutsuna.

**Data curation:** Hideharu Hagiya.

**Formal analysis:** Hideharu Hagiya.

**Funding acquisition:** Satoshi Kutsuna.

**Investigation:** Naruto Kamada.

**Methodology:** Hideharu Hagiya, Satoshi Kutsuna.

**Project administration:** Hideharu Hagiya.

**Software:** Hideharu Hagiya.

**Supervision:** Satoshi Kutsuna.

**Visualization:** Hideharu Hagiya.

**Writing – original draft:** Naruto Kamada.

**Writing – review & editing:** Hideharu Hagiya, Satoshi Kutsuna.

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
