## [Decision Letter · Decision Letter 0]

22 Apr 2025

PONE-D-25-11951Impact of COVID-19 on the Awareness and Interest in Infectious Disease Specialization among Japanese Medical StudentsPLOS ONE

Dear Dr. Hagiya,

Thank you for submitting your manuscript to PLOS ONE. After careful consideration, we feel that it has merit but does not fully meet PLOS ONE’s publication criteria as it currently stands. Therefore, we invite you to submit a revised version of the manuscript that addresses the points raised during the review process.

We look forward to receiving your revised manuscript.

Kind regards,

Yoshito Nishimura, MD, PhD, MPH

Academic Editor

PLOS ONE

Journal Requirements:

This work was conducted as part of "The Nippon Foundation - Osaka University Project for Infectious Disease Prevention".

None

This work was conducted as part of "The Nippon Foundation - Osaka University Project for Infectious Disease Prevention".

This work was conducted as part of "The Nippon Foundation - Osaka University Project for Infectious Disease Prevention".

5. Please remove all personal information, ensure that the data shared are in accordance with participant consent, and re-upload a fully anonymized data set.

Additional Editor Comments :

Thank you for the opportunity to handle the manuscript. While the manuscript contains some potential academic contributions to the field, there are limitations and edits required as suggested by the reviewers. Please address the reviewer comments at the best of your discretion.

Reviewers' comments:

Reviewer's Responses to Questions

**Comments to the Author**

1. Is the manuscript technically sound, and do the data support the conclusions?

Reviewer #1: Yes

Reviewer #2: Yes

Reviewer #3: Yes

2. Has the statistical analysis been performed appropriately and rigorously? 

Reviewer #1: Yes

Reviewer #2: Yes

Reviewer #3: Yes

3. Have the authors made all data underlying the findings in their manuscript fully available?

Reviewer #1: Yes

Reviewer #2: Yes

Reviewer #3: Yes

4. Is the manuscript presented in an intelligible fashion and written in standard English?

Reviewer #1: Yes

Reviewer #2: Yes

Reviewer #3: Yes

5. Review Comments to the Author

Reviewer #1: In the paper “Impact of COVID-19 on the Awareness and Interest in Infectious Disease Specialization among Japanese Medical Students,” the authors conclude underscore the necessity of enhanced educational initiatives to promote ID specialization among medical students, addressing current shortages and future infectious disease preparednessThis study is important for fostering specialization in infectious diseases. However, it lacks the methods, and discussion sections. Therefore, I recommend adding these elements. Thank you for the opportunity to conduct a peer review.

Some suggestions for major revisions are as follows:

1. The abstract includes “Methods: This nationwide cross-sectional survey,” but the main text does not. The description of the methods is inconsistent, so please add it for clarity.

2. Discussion: The paragraph starting with “This drastic shift in educational~” on p. 17 describes differences in perceptions by country. However, differences in medical education and cultural background should also be considered.

Some suggestions for minor revisions are as follows:

1. P9 Questionaries: Please add a statement about the validity of the questionnaire, including references to previous studies, discussions among researchers, and other relevant sources.

2. P11 Results: Please state the number of distributed questionnaires and the response rate.

Reviewer #2: Thank you for the opportunity to review this manuscript. While the topic addressed in the study is significant, there are several limitations in the research that should be considered.

Study Design: The research employs a cross-sectional survey method, which limits the ability to establish causal relationships. Since the goal of the study is to explore the impact of the pandemic on students' career aspirations, a longitudinal approach would have been more suitable for drawing stronger conclusions about the long-term effects.

Narrow Scope: The survey was conducted exclusively among Japanese medical students, which restricts the generalizability of the findings. Including a more diverse, international sample could have strengthened the conclusions and provided a broader understanding of how the pandemic has influenced medical students worldwide, particularly regarding their interest in infectious diseases.

Sampling Bias: The survey, which gathered responses from 502 medical students, shows a disproportionate representation of upper-year students. This bias may skew the results and raise concerns about the sample's representativeness, making it challenging to apply the findings to the broader population of medical students across Japan.

Consequently, the study does not provide actionable recommendations for improving medical education or policy. Offering more concrete suggestions on how to enhance educational programs or address the shortage of infectious disease specialists would have provided valuable insights for both readers and policymakers.

Reviewer #3: I appreciate the authors' efforts in conducting a survey that offers valuable insights into Japanese medical students’ awareness of and interest in infectious disease (ID) specialization. The study highlights increasing awareness but also underscores the persistent challenge of attracting students to the ID field. While the work addresses a critical workforce issue, further clarification is needed in several areas.

• As the authors note in the limitations section, only about 1% of medical students in Japan responded. Therefore, the generalizability of the findings is a major concern. It would be helpful to provide more detailed contextual information—such as the total number of medical schools and medical students in Japan.

• Is the Japan Association for Medical Student Societies an organization that includes all Japanese medical students, or is membership voluntary? More detailed background on this organization is needed.

• The survey distribution method (via Google Forms) should be described more clearly and transparently. How exactly was the survey disseminated?

• Why did 21.5% of all responses come from a single medical school, particularly third-year students? This may have biased the findings and deserves further explanation.

• Whether a respondent’s medical school has an infectious diseases department could be an important factor influencing awareness and interest. Was this information collected? If not, the authors might check this information and consider analyzing how institutional presence or absence of an ID department could affect student perceptions.

• The authors discuss the shortage of ID specialists in Japan and frame the issue around student awareness. However, it would be valuable to describe the underlying reasons for this shortage in the Japanese context. In the United States, for example, common reasons include low compensation, high workload, burnout, and limited exposure during training. Providing comparable context for Japan would help international readers better understand the challenges.

Minor comments

• Figures 1–4: Please include the actual number of responses (n) in addition to proportions.

• Line 75: The number of ID specialists may need to be updated, as the data cited is from October 2024, more than six months ago.

6. PLOS authors have the option to publish the peer review history of their article (what does this mean?). If published, this will include your full peer review and any attached files.

Reviewer #1: No

Reviewer #2: No

Reviewer #3: No

---

## [Author Response · Author response to Decision Letter 1]

1 Jun 2025

Manuscript reference: PONE-D-25-11951-R1

Impact of COVID-19 on the Awareness and Interest in Infectious Disease Specialization among Japanese Medical Students

PLOS ONE

Editor-in-Chief

We would like to thank you for the opportunity to resubmit to you a revised version of our manuscript. We also would like to take this opportunity to express our thanks to the reviewers for their valuable feedback and helpful comments. We have carefully considered your comments and addressed them as thoroughly as possible. Point-by-point responses to the reviewers’ comments are given below. The changes in the manuscript are highlighted.

Based on the journal recommendations, we would like to change as follows;

Competing interests

The authors have declared that no competing interests exist.

Funding

This work was conducted as part of "The Nippon Foundation - Osaka University Project for Infectious Disease Prevention". The funders had no role in study design, data collection and analysis, decision to publish, or preparation of the manuscript.

We hope that the revised manuscript is acceptable for publication in your esteemed journal. We would be glad to respond to any further questions and comments that you may have. Thank you for your consideration.

Sincerely,

Hideharu HAGIYA, M.D., Ph.D., CTH, CIC

Associate Professor, Department of Infectious Diseases, Okayama University Hospital

Mailing Address: 2-5-1 Shikata-cho, Kitaku, Okayama 700-8558, Japan

Review Comments to the Author

Reviewer #1

In the paper “Impact of COVID-19 on the Awareness and Interest in Infectious Disease Specialization among Japanese Medical Students,” the authors conclude underscore the necessity of enhanced educational initiatives to promote ID specialization among medical students, addressing current shortages and future infectious disease preparedness. This study is important for fostering specialization in infectious diseases. However, it lacks the methods, and discussion sections. Therefore, I recommend adding these elements. Thank you for the opportunity to conduct a peer review.

Some suggestions for major revisions are as follows:

1. The abstract includes “Methods: This nationwide cross-sectional survey,” but the main text does not. The description of the methods is inconsistent, so please add it for clarity.

Response

We have added “nationwide” in the method section of the main text.

2. Discussion: The paragraph starting with “This drastic shift in educational~” on p. 17 describes differences in perceptions by country. However, differences in medical education and cultural background should also be considered.

Response

According to your comment, we have added the following sentence in the discussion section; “Due to the inherent heterogeneity in medical education system and sociocultural contexts across global regions, direct comparative analysis presents methodological challenges.” (Line 237-239)

Some suggestions for minor revisions are as follows:

1. P9 Questionaries: Please add a statement about the validity of the questionnaire, including references to previous studies, discussions among researchers, and other relevant sources.

Response

The following sentence was added “The questionnaire items were methodically formulated and refined through discussions among the principal investigators of the study.” (Line 128-129)

2. P11 Results: Please state the number of distributed questionnaires and the response rate.

Response

The questionnaire was disseminated to medical students by researchers through the aid of the Japan Association for Medical Student Societies. The distribution methodologies varied according to individual institutional members, thereby precluding accurate quantification of the total survey instruments distributed and the subsequent response rate. This methodological constraint represents a significant limitation of the current investigation and has been appropriately acknowledged in the limitations section of the manuscript. (Line 306-308)

Reviewer #2

Thank you for the opportunity to review this manuscript. While the topic addressed in the study is significant, there are several limitations in the research that should be considered.

Study Design: The research employs a cross-sectional survey method, which limits the ability to establish causal relationships. Since the goal of the study is to explore the impact of the pandemic on students' career aspirations, a longitudinal approach would have been more suitable for drawing stronger conclusions about the long-term effects.

Response

We appreciate your comments. Based on your suggestions, we have added the following sentences in the limitation section; “Finally, due to the inherent limitations of the cross-sectional study design, a causal relationship between the COVID-19 pandemic and the observed changes in students' career aspirations cannot be definitively established. A longitudinal cohort study design would have potentially yielded more robust conclusions regarding temporal associations and causal mechanisms” (Line 320-324)

Narrow Scope: The survey was conducted exclusively among Japanese medical students, which restricts the generalizability of the findings. Including a more diverse, international sample could have strengthened the conclusions and provided a broader understanding of how the pandemic has influenced medical students worldwide, particularly regarding their interest in infectious diseases.

Response

Your suggestion is appreciated and academically pertinent. Multinational collaborative investigations would indeed provide more methodologically robust conclusions regarding the impact of the COVID-19 pandemic on medical students' future career trajectories. However, the primary objective of the current investigation was specifically to elucidate the situation within the Japanese healthcare education context. We have incorporated your valuable recommendation into the manuscript as a direction for future research initiatives as follows:

“Future investigations should incorporate a multinational perspective, which would enhance our understanding of the relationship between pandemic experiences and medical students' career development, thereby contributing to the international academic community.” (Line 335-339)

Sampling Bias: The survey, which gathered responses from 502 medical students, shows a disproportionate representation of upper-year students. This bias may skew the results and raise concerns about the sample's representativeness, making it challenging to apply the findings to the broader population of medical students across Japan.

Response

We have added the potential selection bias in the limitation section as follows: “Also, the disproportionate representation across academic years potentially introduced selection bias that may have influenced the results. ” (Line 310-311)

Reviewer #3

I appreciate the authors' efforts in conducting a survey that offers valuable insights into Japanese medical students’ awareness of and interest in infectious disease (ID) specialization. The study highlights increasing awareness but also underscores the persistent challenge of attracting students to the ID field. While the work addresses a critical workforce issue, further clarification is needed in several areas.

• As the authors note in the limitations section, only about 1% of medical students in Japan responded. Therefore, the generalizability of the findings is a major concern. It would be helpful to provide more detailed contextual information—such as the total number of medical schools and medical students in Japan.

Response

We have added the background information in the study design section as follows;

At the time of study implementation, Japan's medical education landscape comprised 82 medical schools (42 national, 8 public, 31 private, and 1 national defense), with a collective enrollment of approximately 56,000 medical students. (Line 99-102)

• Is the Japan Association for Medical Student Societies an organization that includes all Japanese medical students, or is membership voluntary? More detailed background on this organization is needed.

Response

We have provided additional information on the society as follows;

A web-based questionnaire created using Google Forms was distributed to medical students by the researchers through the aid of the Japan Association for Medical Student Societies, a voluntary membership organization mainly run by medical students themselves. (Line 95-98)

• The survey distribution method (via Google Forms) should be described more clearly and transparently. How exactly was the survey disseminated?

Response

Thank you for the indication. The sentence was revised as follows;

A web-based questionnaire created using Google Forms was distributed to medical students by the researchers through the aid of the Japan Association for Medical Student Societies, a voluntary membership organization mainly run by medical students themselves. Distribution methodologies were institution-specific, depending on the members of each institution. (Line 95-99)

• Why did 21.5% of all responses come from a single medical school, particularly third-year students? This may have biased the findings and deserves further explanation.

Response

The precise reason for such a disproportionate response distribution remained indeterminate even to the investigative team. This selection bias is explicitly addressed in the Limitations section of the manuscript. (Line 312-316)

• Whether a respondent’s medical school has an infectious diseases department could be an important factor influencing awareness and interest. Was this information collected? If not, the authors might check this information and consider analyzing how institutional presence or absence of an ID department could affect student perceptions.

Response

Your input is greatly appreciated. As noted, a potential correlation may exist between the institutional presence of infectious diseases departments and student response patterns. However, this demographic variable was not incorporated into our data collection methodology, thereby precluding further analytical investigation regarding how the organizational presence or absence of an infectious diseases department might influence student perceptions and attitudes. (Line 313-316)

• The authors discuss the shortage of ID specialists in Japan and frame the issue around student awareness. However, it would be valuable to describe the underlying reasons for this shortage in the Japanese context. In the United States, for example, common reasons include low compensation, high workload, burnout, and limited exposure during training. Providing comparable context for Japan would help international readers better understand the challenges.

Response

Thank you for your input. We developed an additional paragraph explaining the Japanese situation related to ID specialty career development. (Line 281-292)

Minor comments

• Figures 1–4: Please include the actual number of responses (n) in addition to proportions.

Response

All the necessary numbers were additionally given in the figure legend or figure itself.

• Line 75: The number of ID specialists may need to be updated, as the data cited is from October 2024, more than six months ago.

Response

The data was appropriately updated.

---

## [Decision Letter · Decision Letter 1]

17 Jul 2025

Impact of COVID-19 on the Awareness and Interest in Infectious Disease Specialization among Japanese Medical Students

PONE-D-25-11951R1

Dear Dr. Hagiya,

We’re pleased to inform you that your manuscript has been judged scientifically suitable for publication and will be formally accepted for publication once it meets all outstanding technical requirements.

Kind regards,

Yoshito Nishimura, MD, PhD, MPH

Academic Editor

PLOS ONE

Additional Editor Comments (optional):

Reviewers' comments:

Reviewer's Responses to Questions

**Comments to the Author**

1. If the authors have adequately addressed your comments raised in a previous round of review and you feel that this manuscript is now acceptable for publication, you may indicate that here to bypass the “Comments to the Author” section, enter your conflict of interest statement in the “Confidential to Editor” section, and submit your "Accept" recommendation.

Reviewer #1: All comments have been addressed

Reviewer #3: All comments have been addressed

2. Is the manuscript technically sound, and do the data support the conclusions?

Reviewer #1: Yes

Reviewer #3: Yes

3. Has the statistical analysis been performed appropriately and rigorously? 

Reviewer #1: Yes

Reviewer #3: Yes

4. Have the authors made all data underlying the findings in their manuscript fully available?

Reviewer #1: Yes

Reviewer #3: Yes

5. Is the manuscript presented in an intelligible fashion and written in standard English?

Reviewer #1: Yes

Reviewer #3: Yes

6. Review Comments to the Author

Reviewer #1: This study is important for fostering specialization in infectious diseases.

Thank you for the polite correction to the comment. I have no additional comments.

Reviewer #3: I think the authors have addressed all my previous comments. The revisions have improved the manuscript, and I have no further concerns. I believe this work will be helpful for readers not only in Japan but also internationally.

7. PLOS authors have the option to publish the peer review history of their article (what does this mean?). If published, this will include your full peer review and any attached files.

Reviewer #1: No

Reviewer #3: No

---

## [Editor Report · Acceptance letter]

PONE-D-25-11951R1

PLOS ONE

Dear Dr. Hagiya,

I'm pleased to inform you that your manuscript has been deemed suitable for publication in PLOS ONE. Congratulations! Your manuscript is now being handed over to our production team.

Kind regards,

on behalf of

Dr. Yoshito Nishimura

Academic Editor

PLOS ONE